# Infectious Spleen and Kidney Necrosis Virus (ISKNV) Triggers Mitochondria-Mediated Dynamic Interaction Signals via an Imbalance of Bax/Bak over Bcl-2/Bcl-xL in Fish Cells

**DOI:** 10.3390/v14050922

**Published:** 2022-04-28

**Authors:** Pin-Han Chen, Tsai-Ching Hsueh, Jen-Leih Wu, Jiann-Ruey Hong

**Affiliations:** 1Lab of Molecular Virology and Biotechnology, Department of Biotechnology and Bioindustry Sciences, Institute of Biotechnology, National Cheng Kung University, No. 1. University Road, Tainan City 701, Taiwan; betty01269@gmail.com (P.-H.C.); winks0810@gmail.com (T.-C.H.); 2Institute of Biotechnology, National Cheng Kung University, No. 1. University Road, Tainan City 701, Taiwan; 3Laboratory of Marine Molecular Biology and Biotechnology, Institute of Cellular and Organismic Biology, Academia Sinica, Nankang, Taipei 115, Taiwan; jlwu@gate.sinica.edu.tw

**Keywords:** apoptosis, Bcl-2/Bcl-xL, Bax/Bak, DNA virus, host cells, mitochondria, iridovirus

## Abstract

The molecular pathogenesis of infectious spleen and kidney necrosis virus (ISKNV) infections is important but has rarely been studied in connection to host organelle behavior. In the present study, we demonstrated that ISKNV can induce host cell death via a pro-apoptotic Bcl-2 and anti-apoptotic Bcl-2 family member imbalance in mitochondrial membrane potential (MMP or ΔΨm) regulation in GF-1 cells. The results of our study on ISKNV infection showed that it can induce host cell death by up to 80% at day 5 post-infection. Subsequently, in an apoptotic assay, ISKNV infection was seen to induce an increase in Annexin-V-positive signals by 20% and in propidium iodide (PI) staining-positive signals by up to 30% at day 5 (D5) in GF-1 cells. Then, through our studies on the mechanism of cell death in mitochondria function, we found that ISKNV can induce MMP loss by up to 58% and 78% at days 4 and 5 with a JC1 dye staining assay. Furthermore, we found that pro-apoptotic members Bax and Bak were upregulated from the early replication stage (day one) to the late stage (day 5), but the expression profiles were very dynamically different. On the other hand, by Western blotted analysis, the anti-apoptotic members Bcl-2 and Bcl-xL were upregulated very quickly at the same time from day one (two-fold) and continued to maintain this level at day five. Finally, we found that pro-apoptotic death signals strongly activated the downstream signals of caspase-9 and -3. Taken together, these results suggest that ISKNV infection can induce Bax/Bak-mediated cell death signaling downstream of caspase-9 and -3 activation. During the viral replication cycle with the cell death induction process, the anti-apoptotic members Bcl-2/Bcl-xL interacted with the pro-apoptotic members Bax/Bak to maintain the mitochondrial function in the dynamic interaction so as to maintain the MMP in GF-1 cells. These findings may provide insights into DNA-virus control and treatment.

## 1. Introduction

The iridovirus has caused a serious decline in the global population of amphibians and heavy economic losses in the aquaculture field [1,2]. The iridovirus belongs to a large dsDNA virus family that displays icosahedral symmetry with a 120–200 nm diameter. The family Iridoviridae is currently composed of five genera: Chloriridovirus, Iridovirus, Lymphocystivirus, Ranavirus, and Megalocytivirus [3]. Up to now, the megalocytiviruses are largely divided into the RSIV, ISKNV and TRBIV groups, which named as a new member, the scaledrop iridovirus. The megalocytiviruses infect a wide range of freshwater fish and tropical marine fish including groupers, sea bass, gourami, cichlids, red sea bream, angel fish, and lamp eyes, and cause similar diseases in each species, which also induce a lot of mortality and produce heavy economic losses in the aquaculture industry [1,2,3]. During such infections in host fish, Megalocytivirus is also found in some important target tissues such as the spleen, kidney, and gastrointestinal tract, and to lesser extent in the liver, heart, gills, and connective tissues [4,5], which can cause high rates of mortality in fish of economic importance [1,2,3]. Thus, understanding the pathogenesis of ISKNV infection with a focus on new ways of developing effective prevention strategies against the virus infection will also be required in the future.

Multicellular organisms undergo systematic self-destruction through apoptosis in induvial cell responses to a wide variety of stimuli [6]. Therefore, apoptosis plays a central role in the normal development and homeostasis of multicellular organisms [7] and the pathogenesis of viral diseases [8].

The cells’ fate to undergo apoptosis depends mainly on the dynamic balance between the Bcl-2 family sensor proteins, which both promote and inhibit apoptosis. Members of the Bcl-2 family of sensor proteins, such as Bcl-2, Bcl-xl Bax, Bad, Bid, and Bak, represent key points in the apoptotic pathways. These proteins appear to sit at nodes in the apoptotic pathways at points of integration for the stimuli that provoke apoptosis. The Bcl-2-family proteins appear to influence the activation of caspase-family members, which perform the late phase of apoptosis by cleaving a number of cellular proteins and bringing about the destruction of cellular structures [9,10,11].

Mitochondrial membrane permeabilization can alter both the outer and inner mitochondria membranes, and these changes can precede the signs of necrotic or apoptotic cell death and the activation of apoptosis-specific caspases [12,13,14]. Many proteins from the Bcl-2/Bax family regulate apoptosis when they are present in the mitochondrial membrane. The homologues of Bcl-2, including Bcl-xL, Bcl-W, Mcl-1, and A1, reside in the mitochondria, and they serve to stabilize the barrier function of the mitochondrial membranes [15]. Pro-apoptotic proteins such as Bax, Bad, and Bid can shuttle between mitochondrial and non-mitochondrial localizations in the cytosols, which regulate permeabilization via insertion into the mitochondrial membranes when apoptosis is induced [16,17]. 

Recently, Chinese giant salamander iridovirus (GSIV) infection has been shown to induce host cell apoptosis through the upregulation of the pro-apoptotic death genes Bax and Bak at the middle replication stage, and factors in the grouper fin cell line (GF-1) have been shown to modulate this process. In studying the mechanism of cell death, we found that the upregulated, de novo-synthesized Bax and Bak proteins formed heterodimers. This upregulation process correlated with a mitochondrial membrane potential (MMP) loss, increased caspase-3 activity, and increased apoptotic cell death [5,18]. Recently, iridoviruses have been shown to induce cell death including typical apoptotic and non-apoptotic cell death [5]. Subsequently, more advances have been made on GSIV serine/therione kinase-induced cell death, which is blocked by Bcl-2/Bcl-xL overexpression in GF-1 cells [19]. Under the Megalocytivirus system, especially the ISKNV strain, the mechanisms of host cell death have been rarely studied.

Recently, evidence has shown that ISKNV-like (RSIV-KU; Accession number: KT781098) strain [20] infection can induce host cell death. Studying the mechanisms and physiologic functions of cell death during such virus infections can contribute to a better understanding of the host–pathogen interactions and the disease pathogenesis, which could lead to new therapeutic strategies for dealing with infectious diseases [21,22,23,24,25]. Therefore, a detailed elucidation of the mechanism for ISKNV infection-induced cell death is urgently needed.

ISKNV infection induces molecular cell death mechanisms and is correlated to host organelles and applied its on viral controlling, which has been little studied. In the present study, we demonstrated that ISKNV induces apoptotic cell death via a Bax/Bak upregulation for interaction with other Bcl-2 members such as Bcl-2 and Bcl-xL in the mitochondria. This molecular interaction is correlated to the expression level and its downstream death signaling strength by caspase-9/caspase-3 cleavage in host GF-1 cells. These findings may provide new insights into megalocytivirus-induced pathogenesis and control strategies.

## 2. Materials and Methods

### 2.1. Chemicals, Drugs and Antibodies Used

The experiments required the use of a MitoCapture reagent (Mitochondria BioAssayTM Kit; BioVision, Mountain View, CA, USA) and an Annexin V-fluorescein assay (Boehringer-Mannheim, Mannheim, Germany). The following antibodies were used: anti-rabbit Bax and Bak (BD Biosciences, Taipei, Taiwan), anti-Bcl-2, Bcl-xL (BD Biosciences, Taipei, Taiwan), anti-mouse β-actin (Millipore, Darmstadt, Germany), anti-mouse caspase-3 (Calbiochem, Darmstadt, Germany), and anti-mouse caspase-9 (Cell Signaling, Danvers, MA, USA).

### 2.2. Cell Culture and Virus

Grouper fin cells (GF-1 cells) were provided by Dr. Chi (Institute of Zoology and Development of Life Sciences, Taipei, Taiwan). The cells were maintained under standard conditions (at 28 °C) in Leibovitz’s L-15 medium, supplemented by 5% fetal bovine serum (FBS) and 25 µg/mL of gentamycin antibiotic. The cells were passaged twice weekly. All the experiments were performed with cells in the logarithmic growth phase. Naturally infected Pagrus major (red seabream) were collected in 2016 in the Kaohsiung Prefecture, and these fish were the source of the RSIV-KU that was used to infect the GF-1 cells used in this study. RSIV-KU is an ISKNV-like strain, with complete viral genome sequencing [20]. The virus was purified and stored at −80 °C until use. The viral titer was determined by using the median tissue culture infective dose (TCID_50_) assay, in accordance with the method developed by Dobos et al. [26].

### 2.3. Cell Viability

The GF-1 cells were seeded at 1 × 10^5^ cells per ml on a 60 mm Petri dish for at least 20 h prior to cultivation. Then, the resulting monolayers were rinsed twice with PBS, after which the cells were infected with virus at an MOI of 1, and then incubated for 0, 1, 2, 3, 4, and 5 d. Cell viability was determined in triplicate using a trypan blue dye exclusion assay [27]. The viability data are the mean of three independent experiments ±SEM and were analyzed using either paired or unpaired Student’s *t*-tests as appropriate. A value of *p* < 0.05 was taken to represent a statistically significant difference between the mean values of the groups.

### 2.4. Annexin V-FLUOS Staining

An analysis of PS on the outer leaflet of the apoptotic cell membranes was performed using annexin V-fluorescein and propidium iodide (PI) to differentiate apoptotic from necrotic cells. At the end of various incubation times (0, 1, 2, 3, 4, and 5 d), each sample was removed from the medium and washed with PBS, and then cells were incubated with 100 μL of staining solution (annexin V-fluorescein in a HEPES buffer containing PI; Boehringer-Mannheim, Mannheim, Germany) for 10–15 min. Evaluation was performed by fluorescence microscopy using a 488 nm excitation and a 515 nm long-pass filter for detection [28].

### 2.5. Evaluation of Mitochondrial Membrane Potential Using a Lipophilic Cationic Dye

Changes in the mitochondrial membrane potential occurring during apoptosis induced by ISKNV were examined using a JC1 Mito-ID membrane potential aggregation dye. The GF-1 cells were seeded at 1 × 10^5^ cells per ml on a 60 mm Petri dish for at least 20 h prior to cultivation. Then, the resulting monolayers were rinsed twice with PBS, after which the cells were infected with virus at a MOI of 1, and then incubated for 0, 1, 2, 3, 4, and 5 d. At different time points after infection, the cells were washed, fixed, and permeabilized with PBS containing 0.2% Triton X-100-PBS for 5 min on ice. Then, they were incubated with Mito-id JC1 dye for 30 min at 28 °C, washed with PBS, suspended in LB medium, and analyzed by fluorescence microscopy [22,29]. Evaluation was performed by fluorescence microscopy using a 488 nm excitation and a 515 nm long-pass filter for detection.

### 2.6. Western Blotting Analysis

The GF-1 cells were seeded at 1 × 10^5^ cells per ml on a 60-mm Petri dish for at least 20 h prior to cultivation. Then, the resulting monolayers were rinsed twice with PBS, after which the cells were infected with virus at a MOI of 1, and then incubated for 0, 1, 2, 3, 4 and 5 d. The resulting monolayers were rinsed twice with PBS, after which the cells were infected with virus at a MOI of 1, and then incubated for 0 to 5 d. The culture media were aspirated at the end of each time point, after which the cells were washed with PBS and lysed in 0.3 mL lysis buffer (10 mM Tris base, 20% glycerol, 10 mM sodium dodecyl sulfate and 2% ß-mercaptoethanol; pH 6.8). Proteins were separated by sodium dodecyl sulfate polyacrylamide gel electrophoresis, electro blotted and then subjected to immunodetection, as previously described [28]. The membranes were incubated with a dilution of anti-mouse Bax MAb (1:1000; BD Biosciences, San Jose, CA, USA) [22], anti-mouse Bak antibody (BD Biosciences) [22], anti-mouse Bcl-2 antibody (BD Biosciences) [18] anti-mouse Bcl-xL antibody (BD Biosciences) [18], anti-mouse caspase-3 MAb (1:1000, BD Biosciences), anti-mouse caspase-9, anti-mouse ß-actin MAb (1:2,500; Chemicon, Temecula, CA, USA), and a dilution of an appropriate secondary antibody (1:7500 to 1:10,000) that included peroxidase-labeled goat anti-mouse (Amersham, Piscataway, NJ, USA) and goat anti-rabbit (Amersham) antibodies. Chemiluminescence detection was performed using a Western Exposure Chemiluminescence Kit (Amersham), according to the manufacturer’s instructions. The chemoluminescence indication of antibody binding was captured by Top Bio Multigel-21 (Total Lab Systems TLS).

### 2.7. Statistical Analyses

The loss of MM and the percentage of annexin V-fluorescein-positive cells/PI staining were determined in each sample by counting 200 cells. Statistical analyses were performed with SPSS 16.0 or GraphPad Prism 8.2.1 software. Bar plots were expressed as the mean standard deviation of at least three experimental replicates. Differences between two groups were assessed by Student’s *t*-test, and differences between multiple groups were tested by one-way analysis of variance (ANOVA) followed by Tukey’s post hoc test. Bars are expressed as mean ± s.d. or mean ± SEM. Statistical significance is shown at * *p* < 0.05, ** *p* < 0.01.

## 3. Results

### 3.1. ISKNV Induces Host Apoptotic Cell Death in GF-1 Cells 

The examination of ISKNV infection in the cell survival assay showed that it induced host cell damage. In the viability assay, we found that ISKNV can induce cell death from D2 (Figure 1A). Then, we counted the survival rate, as shown in Figure 1B. The results of the ISKNV study showed that it induces host cell death from D2 (26%) to D5 (72%) (Figure 1A,B) compared to D0 (100%) and the mock control (131% (D2) to 167% (D5)). On the other hand, the viral protein MCP was checked and monitored from D0 to D5. In the MCP, the protein was gradually expressed from D1 to D5, with major expression at D3, which correlated to cell death induction.

### 3.2. ISKNV Can Induce Apoptotic Cell Death and Necrosis in GF-1 Cells

Then, we counted the apoptotic cell death/necrosis rate using the Annexin-V/PI staining assay. In the results of the ISKNV-infected groups, we found that at D3, it can apparently induce Annexin-V-positive signals/necrosis (Figure 2A) compared to the D0 group. Figure 2B shows the Annexin-V-positive cells (green florescence) from Figure 2A that increased by up to 10% (D4) and 6% (D5), respectively, compared to the mock group. The PI-positive cells counted for the necrosis assay (red florescence) are shown in Figure 2C and increased by 14% (D4) and 16% (D5), respectively, compared to the mock group.

### 3.3. ISKNV Induces Mitochondrial MMP Loss

To determine whether ISKNV can induce the loss of MMP, mitochondrial function was evaluated using MitoCapture Reagent (JC-1 dye). This dye is trapped in the mitochondria with normal ΔΨm and released from the mitochondria with abnormal ΔΨm into the cytosol, resulting in the loss of fluorescence intensity. In healthy cells, the MitoCapture dye (JC-1) aggregates in the mitochondria and fluoresces red. In apoptotic or secondarily necrotic cells, the dye cannot accumulate in the mitochondria and is distributed throughout the cytoplasm as a fluorescent green monomer. The loss of MMP (Figure 3A; indicated by arrows) was induced by changes in the green or red fluorescence intensity with ISKNV infection at different time points from D2 to D5, which showed losses between 22% and 61%, respectively, compared with the mock groups (Figure 3B).

### 3.4. ISKNV Infection Upregulates Both Anti-Apoptotic Bcl-2-Family Members’ (Bcl-2/Bcl-xL) and Pro-Apoptotic Members’ (Bax/Bak) Expression

Subsequently, we further checked the intrinsic cell death signals with ISKNV infection in GF-1 cells. First, we found that ISKNV infection can upregulate the pro-apoptotic gene Bax very quickly from D1 (Figure 4A, lane 2) to D5 (Figure 4A, lane 6). However, Bak was more slowly upregulated from D2 (Figure 4A, lane 3), showing a gradual increase in upregulation by D5 (Figure 4A, lane 6). These expression levels were counted and are shown in Figure 4B, which shows very different dynamic changes. 

On the other hand, the GF-1 cells with ISKNV infection were also shown to induce preventive signals as anti-apoptotic cell death signals, such as Bcl-2 and Bcl-xL (Figure 5), which shows a very different upregulation pattern. With ISKNV infection, at D1 (Figure 5A, lane 2), downregulation was very quick but slower at D5 (Figure 5A, lane 6). On the other hand, in Bcl-xL, the upregulation pattern was gradually upregulated from D1 (Figure 5A, lane 2), but with a major peak at D3 (Figure 5A, lane 4), which was maintained to D5 (Figure 5A, lane 6). The expressional levels were counted and are shown in Figure 5B, which shows the maintenance of the preventive signals, but why the anti- and pro-apoptotic signals at the same time are strongly maintained is still unknown.

### 3.5. ISKNV Induces Caspase-9 and Caspase-3 Activation at the Replication Stage

Furthermore, we tried to answer the following question: why does the dual upregulation of the positive and negative signals guide cell death? In the downstream signaling of the intrinsic pathway (mitochondria-mediated pathway), caspase-9 and -3 were checked, and we found that two death factors were very quickly upregulated and cleaved from D1 (Figure 6A,B, lane 2) to D5 (Figure 6A,B, lane 6) by Western blotting analysis, which correlates to the Bax/Bak-mediated cell death pathway shown in Figure 4. 

## 4. Discussion

The megalocytivirus has been insufficiently studied, despite the severe economic losses it causes, especially the ISKNV strain, in a wide range of economically important freshwater and marine fish species in the Asia Pacific region [1,2,3]. In this study, we demonstrated that ISKNV induces host cell death through the upregulation of the pro-apoptotic death factors Bax and Bak from the early replication stage (at D1), which was combated by the anti-apoptotic death members Bcl-2 and Bcl-xL between the early and late replication stages (from D1 to D5) in fish cells. Finally, the mitochondria-mediated Bax/Bak cell death signal can activate the downstream signal of the caspase-9/-3 pathway to trigger host cell death. Therefore, we conclude that this new examination of the ISKNV molecular interaction process could help in elucidating the mechanisms of DNA viral pathogenesis and infection.

### 4.1. ISKNV Induced Annexin-V/PI Positives in GF-1 Cells 

The family Iridoviridae is currently composed of five genera: Ranavirus, Lymphocystivirus, Megalocytivirus, Iridovirus, and Chloriri-dovirus [30]. Although the forms of cell death evoked by iridovirus are continually being disclosed, including typical apoptosis and non-apoptotic cell death [4,31,32,33,34,35,36], the signaling pathways involved in these processes still remain largely unknown.

Recently, iridoviruse has been shown to belong to large-DNA viruses, which can cause high rates of mortality in the global population of amphibians and heavy economic losses in the aquaculture industry [37,38]; invertebrates and lower vertebrates, including crustaceans, mollusks, insects, fish, amphibians, and reptiles, are also affected [3]. In our study, we found that ISKNV can induce host cell death in a viability assay, shown in Figure 1A,B, during ISKNV early expression. Then, we performed further assays using Annexin-V/PI double-positive staining, and confirmed the apoptotic/necrosis process, which revealed the death signals triggered at D2 and D5 post-infection (as shown in Figure 2A–C), and that the apoptotic death signal (Figure 2B; primed at D2) can gradually become necrosis (Figure 2C; primed at D3). The molecular apoptotic pathways require further study.

### 4.2. What Kind of Cell Death Signaling Interaction Occurs during ISKNV-Triggered MMP Loss?

The process of apoptotic cell death is controlled by a diversity of cell-signaling pathways that originate either from the cells’ external environment via an extrinsic pathway by receptor mediators, such as the Fas ligand, or from within the cell itself via intrinsic pathway primed mitochondria-mediated signals, such as cytochrome *c* release [39]. Apoptosis may be used by the host to limit the production of viruses or to limit their replication [39,40]. Thus, viruses limit the apoptosis process to produce sufficient virus progeny or to facilitate virus release in DNA or RNA viruses [39,40,41]. 

Some unique features of eukaryotic cells include the presence of distinct membrane bound structures called organelles. Organelles achieve this due to the presence of an essential set of proteins and a distinctive lipid composition that correlates to their specific function. They are dynamic and interact with the neighboring organelles to regulate their biogenesis and function [42,43,44]. It is interesting that not only are organelles required for the proper functioning of cells but are also required for the successful infection of a virus [45,46]. Recently, viruses have been shown to develop alternate strategies to survive in cells, which correlate with the steps of the viral life cycle including entry, translation, replication, assembly, and egress [47]. Moreover, viruses have developed remarkable ways to complete their life cycle by targeting specific cell organelles and processes. Organelles such as mitochondria, ER, and peroxisomes also play an important role in innate immunity and host defense [48]. In our study, we found that the loss of MMP in mitochondria was induced in ISKNV-infected GF-1 cells (Figure 3), which were primed at D2, followed by a huge induction at D5, up to 60% more than that in the mock group, but this raises the question of why this is the case. 

### 4.3. Are Bcl-2-Family Members Important in ISKN-Induced Cell Death?

A key step in the initiation of intrinsic apoptosis is mitochondrial outer membrane permeabilization (MOMP), which enables the release of pro-apoptotic factors, such as cytochrome *c*, from the mitochondrial intermembrane space [49]. The release of cytochrome *c* to the cytosol triggers the formation of the apoptosome, which induces the activation of initiator caspase-9 downstream [49,50]. Similar to the role of caspase-9, caspase-8 proteolytically activates the effector caspases. BCL2-family proteins serve as important regulators of intrinsic apoptosis by controlling MOMP [51]. The BCL2 family comprises pro- and anti-apoptotic members that share one or more BCL2-homology (BH) domains. Anti-apoptotic proteins (e.g., MCL1 and BCL2) contain four BH domains (BH1–4) and promote cell survival by antagonizing the activity of the proapoptotic BCL2 family members. The pro-apoptotic members can be divided into two subfamilies according to their BH domain composition: multi-domain pro-apoptotic proteins (e.g., BAK and BAX), which contain three BH domains, and ‘BH3-only’ proteins (e.g., BAD, BIK, BID, and PUMA), which contain only the BH3 domain [51,52]. In our study, ISKNV infection was shown to upregulate both pro-apoptotic members Bax/Bak (Figure 4) and anti-apoptotic members Bcl-2/Bcl-xL (Figure 5) at the early replication stage. On the other hand, we found that caspase-9/caspase-3 was activated from precursors between D1 and D5 (Figure 6A,B) and seems to have more unknown signals involved that require further examination. 

On the other hand, Chinese giant salamander iridovirus (GSIV) infection inhibits the expression of AdBcl-xL [53], and overexpression suppresses virus replication. Furthermore, AdBcl-xL inhibits the mitochondrial apoptotic cell death induced by GSIV by binding to Bak, which reduces caspase-3 and caspase-9 activation. Furthermore, polyunsaturated fatty acids (PUFAs) and DHA can upregulate anti-cell death factor Bcl-2 and interact with pro-apoptotic protein Bax in the outer membrane of mitochondria to enhance host viability [25], indicating that Bcl-2 family members are strongly involved, such as pro-apoptotic and anti-apoptotic member interactions for the maintenance of mitochondrial function. 

### 4.4. Does the Viral Genome Provide Cell Death or Anti-Cell Death Signals during the Viral Replication Cycle?

Although some efforts have been made to understand the pathogenesis of ISKNV infection, the interactions between the virus and host are still little understood. ISKNV has a larger genome with a length of 111,362 bp that includes up to 124 potential open reading frames (ORFs), indicating that the relationship between the virus gene and the host response is relatively complicated during viral replication [20,54]. Recently, Grouper iridovirus GIV (Ranavirus strain) has been shown to belong to the Iridoviridae family and harbors GIV66, a putative Bcl-2-like protein, which the potent anti-apoptotic activity of GIV66 by identifying it as a new pro-survival Bcl-2 protein and identifying a pivotal role of pro-apopotic Bim in GIV-mediated inhibition of apoptosis [55]. On the other hand, In GSIV stain, serine/threonine (ST) kinase can induce Bax/mitochondria-mediated cell death pathway, which can be blocked by the interaction of Bcl-xL and Bcl-2 with Bax to inhibit cytochrome *c* release during MMP loss. Furthermore, this rescue activity also correlated with inhibition of caspase-9 and -3 activation, thereby enhancing cell viability and will be considered to apply in viral regulation [56].

## 5. Conclusions

Summary (Figure 7): taken together, our data suggest that ISKNV infection can induce stress signals for the regulation of host cell mitochondrial signaling, either by the upregulation of anti-cell death signals Bcl-2 and Bcl-xL or by the upregulation of pro-cell death signals Bax and Bak from the early to late viral replication stages. Then, stronger death signaling is triggered downstream of caspase-9/-3 activation, and apoptotic-mediated cell death is induced during the replication cycle in GF-1 cells. These findings provided new insights into DNA viruses using a more controlled strategy by which they enhance cell death regulation.

## Figures and Tables

**Figure 1 viruses-14-00922-f001:**
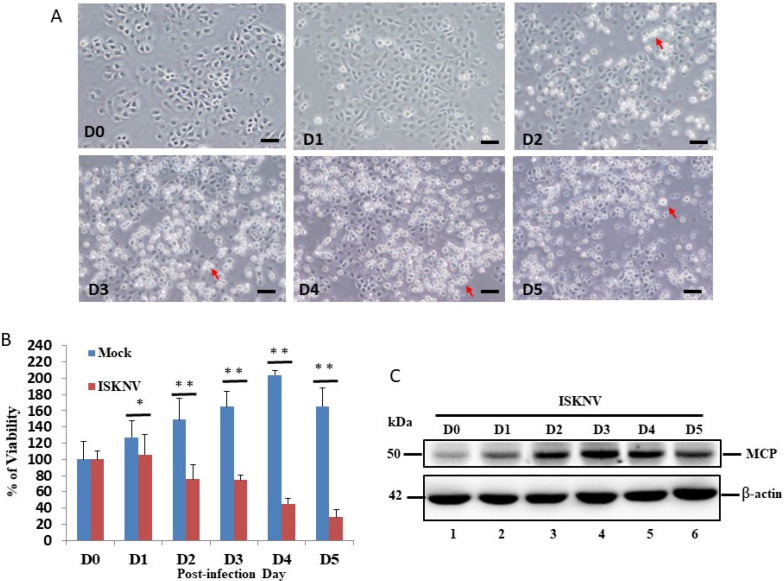
ISKNV infection and expression directly induces cell death in GF-1 cells. (**A**) Phase-contrast micrographs of ISKNV-infected GF-1 cells at different time points (D0-D5). Phase-contrast images in which rounded-up cells are indicated by red arrows. Scale bar = 20 µm. (**B**) The percentage of cell viability of ISKNV-infected cells at different time points. The numbers of virus-infected cells in each 200 cell sample were assessed in three individual experiments. Each point represents the mean ISKNV-infected cells of three independent experiments ± the standard error of the mean (SEM). The data were analyzed using either paired or unpaired Student’s *t*-tests, as appropriate. Statistical significance was defined as * *p*-values < 0.05; ** *p*-values < 0.01. (**C**) Western blot analysis of ISKNV major capsid protein (MCP) expression in GF-1 cells following transfection and incubation for D0 (lane 1), D1 (lane 2), D2 (lane 3), D3 (lane 4), D4 (lane 5), and D5 (lane 6). MCP proteins were detected by Western blot analysis; the gels were immunoblotted with a polyclonal antibody to ISKNV MCP and ß-actin as an internal control.

**Figure 2 viruses-14-00922-f002:**
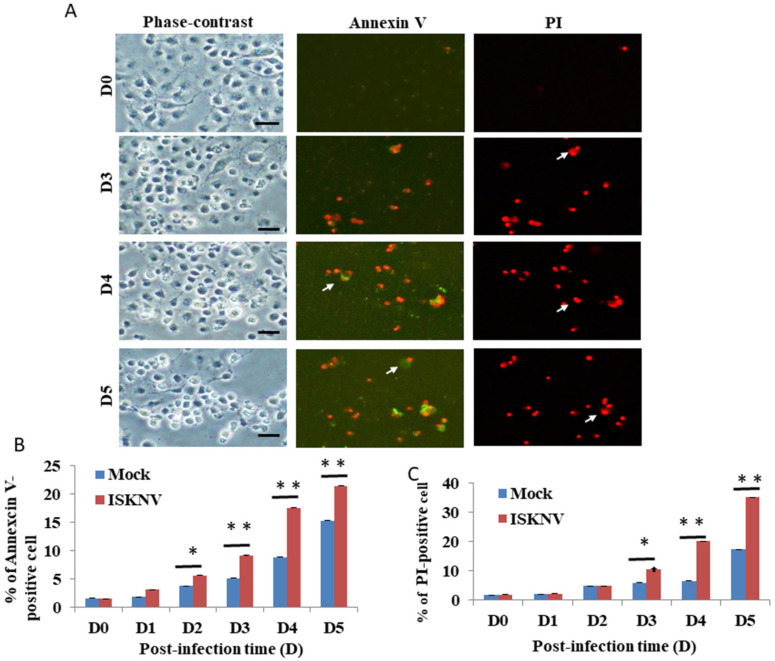
ISKNV directly induces apoptotic/necrosis cell death by dual labeling with Annexin-V/PI staining in GF-1 cells. (**A**) Phase-contrast micrographs of ISKNV-induced apoptotic/necrosis GF-1 cells at different time points (D0, D3, D4, and D5). Phase-contrast (left panels D0, D3 D4, and D5); green fluorescence images for Annexin-V-positive cells (middle panels D0, D3 D4, and D5); and red fluorescence images for necrosis-positive cells (right panels D0, D3 D4, and D5), in which Annexin-V-positive cells and necrosis-positive cells are indicated by arrows. Scale bar = 20 µM. (**B**,**C**) The percentage of apoptotic/necrosis ISKNV-induced cells at different time points. The numbers of ISKNV-infected cells containing apoptotic/necrosis cells in each 200 cell sample were assessed in three individual experiments. Each point represents the mean ISKNV-induced apoptotic/necrosis cells of three independent experiments ± the standard error of the mean (SEM). The data were analyzed using either paired or unpaired Student’s *t*-tests, as appropriate. Statistical significance was defined as * *p*-values < 0.05; ** *p*-values < 0.01.

**Figure 3 viruses-14-00922-f003:**
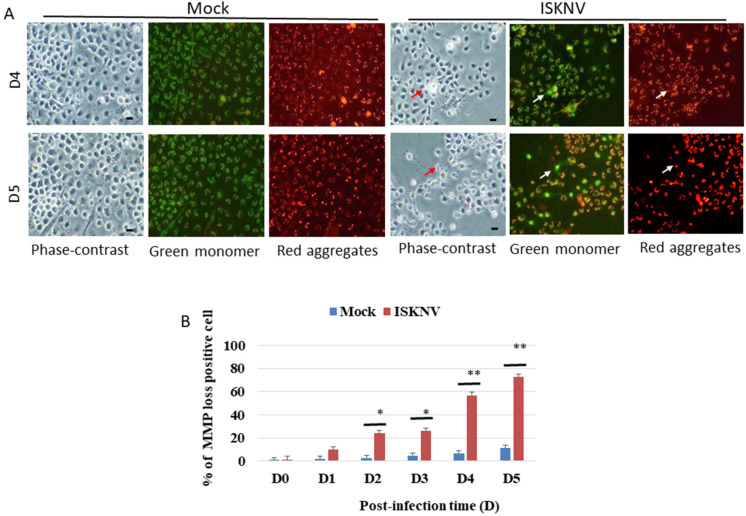
ISKNV infection can induce the loss of MMP in GF-1 cells. (**A**) Identification of ISKNV infection inducing MMP loss in GF-1 cells at D4 and D5 post-infection. Phase-contrast and fluorescence images for mock control group (left panels D4 and D5); green/red fluorescence images (middle panels/right panels) for MMP loss-positive cells. Phase-contrast (ISKNV-infected cells indicated by red arrows) and fluorescence images for ISKNV-infected group (left panels D4 and D5); green/red fluorescence images (middle panels/right panels) for MMP loss-positive cells, in which MMP loss cells are indicated by white arrows. Scale = 10 μm. (**B**) The ISKNV-induced losses of MMP in GF-1 cells were counted at D0, D1, D2, D3, D4, and D5 post-infection. The numbers of MMP-loss cells among each of the 200 cell samples were assessed in three individual experiments. Each point represents the mean MMP-loss cells of three independent experiments ± the standard error of the mean (SEM). The data were analyzed using either paired or unpaired Student’s *t*-tests, as appropriate. Statistical significance was defined as * *p*-values < 0.05; ** *p*-values < 0.01.

**Figure 4 viruses-14-00922-f004:**
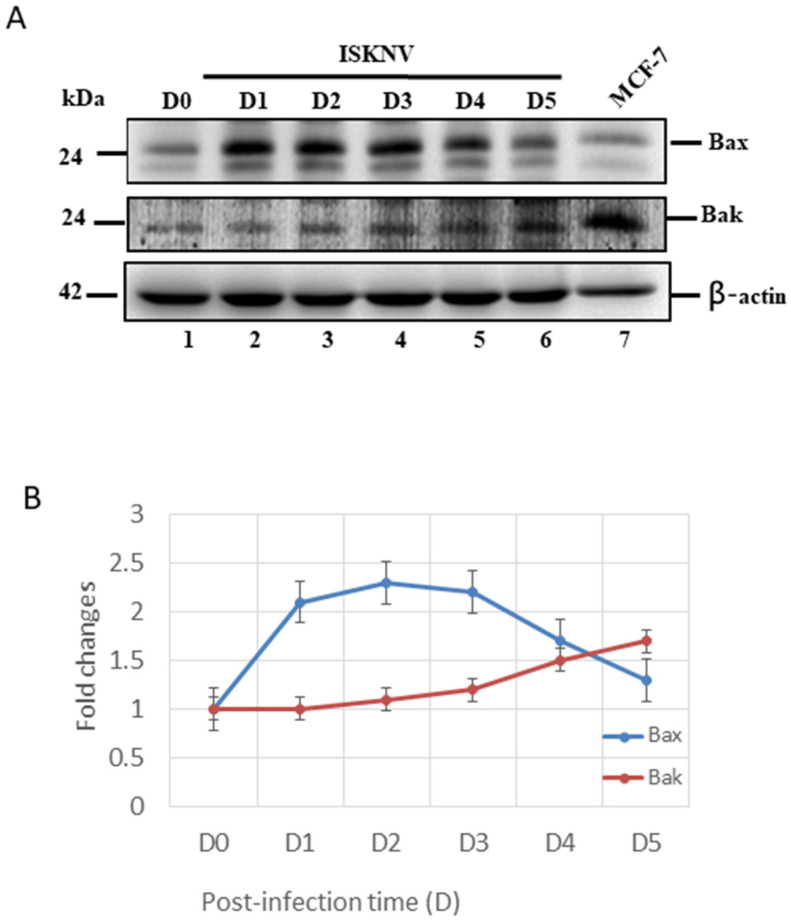
ISKNV infection can upregulate *Bax* and *Bak* gene in GF-1 fish cells. (**A**) Identification of Bax and Bak proteins by Western blot analysis. Infection of GF-1 cells with ISKNV at D0, D1, D2, D3, D4, and D5 pt in GF-1 cells, where lanes 1-6 display ISKNV-infected GF-1 cells; and lane 7 shows the control MCF-7 cell lysate. Blots were probed with an anti-mouse Bax and Bak monoclonal antibody (1:7500), and an anti-mouse β-actin monoclonal antibody (1:12,500). (**B**) Quantification of protein expression level by Image J software, from (**A**) that D0 as a one fold control was counted for compared with other ISKNV-infected groups from D1 to D5, respectively.

**Figure 5 viruses-14-00922-f005:**
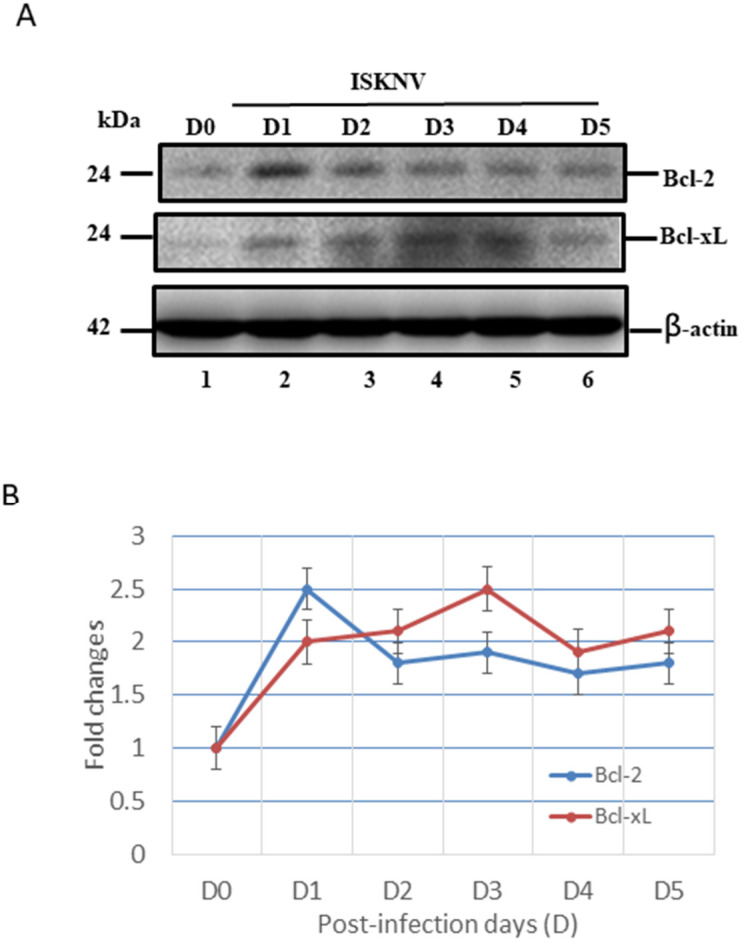
ISKNV infection can upregulate Bcl-2 and Bcl-xL gene in GF-1 fish cells. (**A**) Identification of anti-apoptotic members Bcl-2 and Bcl-xL proteins by Western blot analysis. Infection of GF-1 cells with ISKNV at D0, D1, D2, D3, D4, and D5 pt in GF-1 cells, where lanes 1-6 display ISKNV-infected GF-1 cells. Blots were probed with an anti-mouse Bcl-2 and Bcl-xL monoclonal antibody (1:7000), and an anti-mouse β-actin monoclonal antibody (1:12,500). (**B**) Quantification of protein expression level by Image J software, from (**A**) that D0 as a one fold control was counted for compared with other ISKNV-infected groups from D1 to D5, respectively.

**Figure 6 viruses-14-00922-f006:**
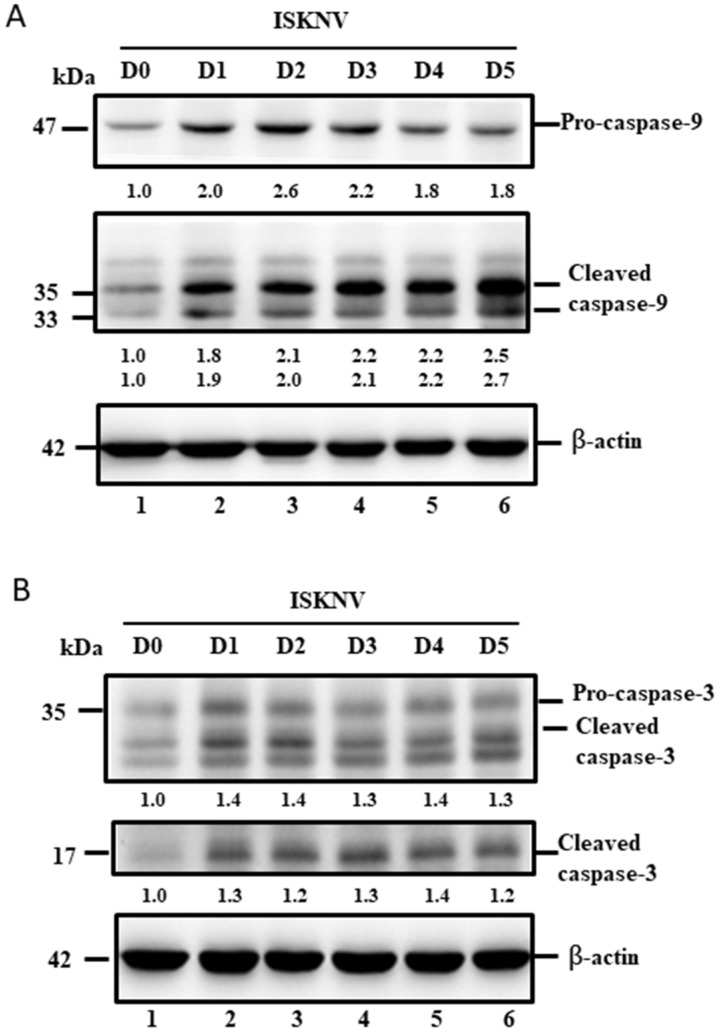
ISKNV infection can activate caspases-9 and -3 in GF-1 cells. (**A**,**B**). Infection of GF-1 cells with ISKNV at D0, D1, D2, D3, D4, and D5 pi in GF-1 cells that showed ISKNV-induced caspase-9 (**A**) and caspase-3 cleavage (**B**) from procaspase-9 and procaspase-3, as determined by Western blot analyses. Lanes 1-6 correspond to ISKNV-infected GF-1 cells at D0, D1, D2, D3, D4, and D5 pt. Blots were probed with an anti-mouse caspase-9 and caspase-3 monoclonal antibody (1:7500) and an anti-mouse β-actin monoclonal antibody (1:12,500). Quantification of protein expression levels by Image J software, from (**A**,**B**) that D0 as a one fold control was counted for compared with other ISKNV-infected groups from D1 to D5, respectively.

**Figure 7 viruses-14-00922-f007:**
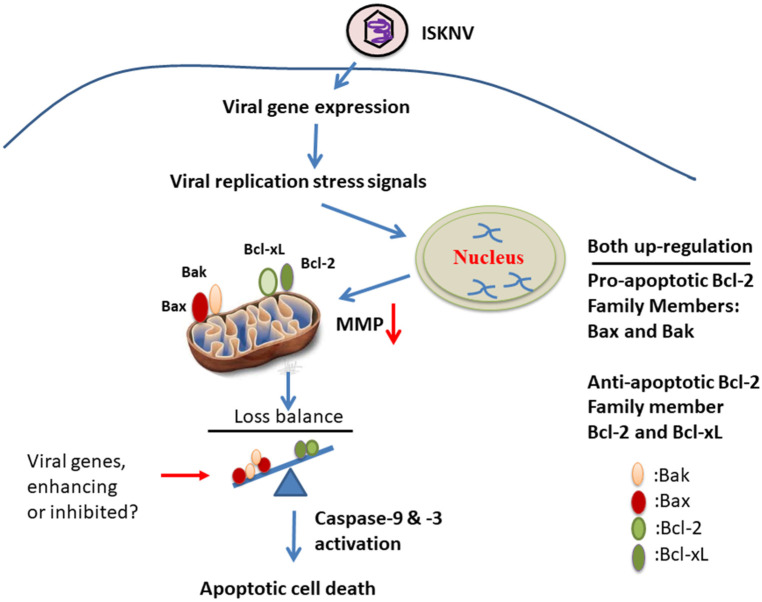
The hypothesis that ISKNV upregulates and induces positive and negative apoptotic cell death cascades on mitochondria function controlling in fish cells. The DNA virus ISKNV shows novel approach using stress signals that can upregulate either anti-apoptotic cell death genes Bcl-2 and Bcl-xL or pro-apoptotic cell death genes Bax and Bak during the viral replication cycle in GF-1 fish cells. This ISKNV-triggered negative and positive cell death signals are also correlated with downstream mitochondrial disruption to MMP loss and caspases-9/-3 activation in GF-1 cells. Interestingly, ISKNV-induced Bax/Bak-mediated cell death signaling might be inhibited or enhanced by viral genes from the viral genome such as GIV66 (apoptosis inhibitor; [55]) and GSIV serine/threonine kinase (apoptosis inducer; [56]) during viral replication.

## Data Availability

The data are contained within the manuscript.

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
