# Peer review of "Infectious Spleen and Kidney Necrosis Virus (ISKNV) Triggers Mitochondria-Mediated Dynamic Interaction Signals via an Imbalance of Bax/Bak over Bcl-2/Bcl-xL in Fish Cells"

_viruses, 2022, doi:10.3390/v14050922_

Round 1
Reviewer 1 Report
Abstract: Box/Bax → Bax/Bak
At the outset, it should be mentioned that megalocytiviruses are largely divided into the RSIV-ISKNV-TRBIV group and a new member, the scaledrop iridovirus.
Although RSIV-KU is described as ISKNV-like, it should be clearly stated whether it is really the closest to ISKNV of the three species in the RSIV-ISKNV-TRBIV group. If it is not, and is closer to RSIV, then the description needs to be corrected, including the title. Since the title says ISKNV, not ISKNV-like, we think it would be a good idea to include the Genbank accession number, if available.
You have not described how the relationship between infection and apoptosis of viruses other than megalocytiviruses, especially those outside the family Iridoviridae, compares to the relationship between ISKNV infection and apoptosis in this study. If it is possible to discuss whether these apoptosis-inducing gene movements are characteristic of megalocytivirus infection or general to many other viral infections, please discuss. If they are characteristic of megalocytivirus infections, then please discuss a little about the relationship of the genes on the viral side that cause the characteristic trends (this is only a possibility, as it is predictable that it may not be clearly understood. For example, the GIV66 gene of Ranavirus or the GSIV ST kinase gene.) These considerations would make this paper more valuable.
The mammalian antibodies used in the study seem to react well with the corresponding antigenic proteins of fish cells, and if there is already evidence in other papers that these antibodies can be used in fish, this should be cited. If this is the first study, the reasons why the antibodies can be used (e.g., the amino acid sequence of the epitope is highly conserved) should be mentioned.
Author Response
Response to Reviewers
In this revised manuscript we taken all the Editor and Reviewers comments and suggests. In the figure legends we have mention D0 as one fold control for comparing with other ISKNV-infected groups from D1 to D5 in Figure 4 and -6. And response to Reviewers as follows:
Reviewer 1:
- Abstract: Box/Bax → Bax/Bak
Ans: In the Abstract have checked the Bax/Bak and corrected them already.
- At the outset, it should be mentioned that megalocytiviruses are largely divided into the RSIV-ISKNV-TRBIV group and a new member, the scaledrop iridovirus.
Ans: Thanks for mention, we have added this information in Introduction. Such as “Up to now, the megalocytiviruses are largely divided into the RSIV, ISKNV and TRBIV groups, which named as a new member, the scaledrop iridovirus.”
- Although RSIV-KU is described as ISKNV-like, it should be clearly stated whether it is really the closest to ISKNV of the three species in the RSIV-ISKNV-TRBIV group. If it is not, and is closer to RSIV, then the description needs to be corrected, including the title. Since the title says ISKNV, not ISKNV-like, we think it would be a good idea to include the Genbank accession number, if available.
Ans: The RSIV-KU stain of Genbank accession number is added and marked in the Introduction. Such as “Recently, evidence has shown that ISKNV-like (RSIV-KU; Accession number: KT781098) strain [21] infection can induce host cell death.”
- You have not described how the relationship between infection and apoptosis of viruses other than megalocytiviruses, especially those outside the family Iridoviridae, compares to the relationship between ISKNV infection and apoptosis in this study. If it is possible to discuss whether these apoptosis-inducing gene movements are characteristic of megalocytivirus infection or general to many other viral infections, please discuss. If they are characteristic of megalocytivirus infections, then please discuss a little about the relationship of the genes on the viral side that cause the characteristic trends (this is only a possibility, as it is predictable that it may not be clearly understood. For example, the GIV66 gene of Ranavirus or the GSIV ST kinase gene.) These considerations would make this paper more valuable.
Ans: In the Discussion section have enhanced to discussion of GIV66 gene of Ranavirus or the GSIV ST kinase gene discovery in recent. Such as :Recently, Grouper iridovirus GIV (Ranavirus strain) has been shown to belong to the Iridoviridae family and harbors GIV66, a putative Bcl-2-like protein, which the potent anti-apoptotic activity of GIV66 by identifying it as a new pro-survival Bcl-2 protein and identifying a pivotal role of pro-apopotic Bim in GIV-mediated inhibition of apoptosis [59]. On the other hand, In GSIV stain, serine/threonine (ST) kinase can induce Bax/mitochondria-mediated cell death pathway, which can be blocked by the interaction of Bcl-xL and Bcl-2 with Bax to inhibit cytochrome c release during MMP loss. Furthermore, this rescue activity also correlated with inhibition of caspase-9 and -3 activation, thereby enhancing cell viability and will be considerly to apply in viral regulation [60].:
- The mammalian antibodies used in the study seem to react well with the corresponding antigenic proteins of fish cells, and if there is already evidence in other papers that these antibodies can be used in fish, this should be cited. If this is the first study, the reasons why the antibodies can be used (e.g., the amino acid sequence of the epitope is highly conserved) should be mentioned.
Ans: In the Human or Mouse system to produce antibody such as Bax/Bak that can crossreact to fish protein, which based on very similar to epitope or conformation. In our system we have tough to screen and checked the commercialized antibodies from mouse or human system. In Bax/Bak has cited in Ref 19; Bcl-2 and Bcl-xL has cited in Ref 30.
Reviewer 2 Report
The present manuscript no. viruses-1685125 investigated molecular pathogenesis of ISKNV and demonstrated pro-apoptotic imbalance in GF-1 cells.
The current study contributes not only to the knowledge of the ISKNV but also its infection in the host cells.
While the work is significant and of interest to researchers working on fish viral disease, there are scope for minor improvement in the manuscript. Overall the manuscript is well written and it is suitable for publication in this journal
Abstract must highlight methodology used in the study.
The significant impact and economic loss from ISKNV to aquaculture industry must be highlighted in the introduction
The clear objectives should be mentioned in the introduction.
For statistical analysis, homogeneity of variance, ANOVA and software used should be mentioned.
Figure 1A. scale bar used for all the image should be mentioned.
Figure 7 could be presented as graphical abstract.
